# UltraMAE: Multi-modal Masked Autoencoder for Ultrasound Pre-training

**Aimon Rahman**    ARAHMA30@JHU.EDU and **Vishal M Patel**    VPATEL36@JHU.EDU
*Department of Electrical and Computer Engineering, Johns Hopkins University*

**Editors:** Accepted for publication at MIDL 2024

## Abstract

Pre-training on a large dataset such as ImageNet followed by supervised fine-tuning has brought success in various deep learning-based tasks. However, the modalities of natural images and ultrasound images have considerable differences, making pre-training on natural images ineffective for ultrasound-related tasks. In this paper, we introduce a unified masking-based model for both ultrasound images and videos that learns better visual representation than the network with single-modality representations. This is the first large-scale generalized ultrasound pre-training network that simultaneously utilizes 100,000+ videos and images of different parts of the human anatomy such as the liver, bones, heart, thyroids, nerves, etc, making the network an effective benchmark pretrained model for any ultrasound-specific downstream tasks. We propose a novel method for ultrasound image analysis that utilizes an ultrasound-specific confidence map to guide low-level representation learning through masked feature acquisition. Our pre-trained network has demonstrated remarkable efficacy and versatility in tackling both classification and segmentation tasks across a range of ultrasound pathologies, highlighting its potential for widespread adoption and impact in the ultrasound field. In addition, we show that our pre-training model can be leveraged to learn efficiently with a small number of labeled ultrasound images.

**Keywords:** Multi-Modal, Ultrasound, Pre-training, Masked Auto Encoder

## 1. Introduction

The cost-effective, portable, radiation-free, and non-invasive nature of ultrasound (US) imaging has made it a popular choice for clinical diagnosis (Hacihaliloglu et al., 2014). Recent advancements in deep learning for medical diagnosis, combined with ultrasound's ability to detect various pathologies with reasonable sensitivity, have created opportunities to develop powerful automated tools for medical diagnosis (Rahman et al., 2022a,b; Wang et al., 2018, 2020). These tools can be leveraged to mitigate sonographers' reading burden and improve efficiency as well as the accuracy of the diagnosis. However, one of the major challenges in developing these kinds of networks is the problem of data scarcity, which is typically dealt with by transferring parameters from pre-trained networks (Azizi et al., 2021). Pre-training on a large natural dataset has been shown to yield a performance boost for the downstream task, however, the significant domain gap between natural and medical images makes pre-training on natural images for downstream medical-related tasks less effective (Chen et al., 2021; Azizi et al., 2021). It's worth noting that ultrasound imaging, compared to other medical imaging modalities, encompasses a wide range of clinical applications and specialties, indicating a significant domain gap between ultrasound and other imaging techniques. Furthermore, the use of different transducers, modes, machines,

and other factors, such as inter-individual variability and transducer pressures, can result in further variability within the same anatomy (Cruz-Lemini et al., 2016). As a result, it becomes imperative to use pre-trained models specifically tailored for ultrasound imaging.

In recent years, there has been a growing trend to leverage unlabelled datasets for pre-training large-scale models in a self-supervised manner, particularly in medical imaging, where labeling can be costly and require domain-specific knowledge (Azizi et al., 2021; Chen et al., 2021; Basu et al., 2022). With the rise of transformer networks, one of the most effective pre-training strategies involves masking out certain pixels in input images and predicting their values, which helps the network to learn robust feature representations (He et al., 2022; Tong et al., 2022). Although recent literature on ultrasound pre-training has demonstrated promising results using various unsupervised techniques for different ultrasound-related tasks (Chen et al., 2021; Basu et al., 2022), most of these studies have focused on pre-training on a single or a few anatomies, such as liver or lungs, and evaluating on similar downstream tasks. Furthermore, these works have not explored the use of masked autoencoder architectures, which have been shown to outperform other self-supervised techniques.

Ultrasound can be in either video or image format. For example, temporal information is necessary for echocardiography, whereas bone scans may only require images. Hence, a generalized pre-trained model that works both for images and videos is necessary for the ultrasound domain. In this paper, we propose **UltraMAE**, the first unified masked pre-training model for ultrasound videos and images. Additionally, instead of predicting the input image, UltraMAE predicts the certainty weighted ultrasound image (CWUS) that enables the model to provide special attention to local details in ultrasound scans. CWUS utilizes a confidence map to selectively enhance the areas of high confidence, resulting in a reduction of noise and shadow artifacts that may accumulate during image acquisition. By training a network to reconstruct CWUS, the model is encouraged to learn from relevant local contexts, leading to improved accuracy and precision in downstream diagnosis. In summary, we present the following contributions in this paper:

- We are the first to propose a simple yet powerful unified masked pre-training model that integrates both ultrasound images and videos into a single framework. This makes the pre-training model scalable to different ultrasound-related downstream tasks with minimal changes.

- We propose a novel framework that aims to reconstruct high-quality ultrasound images/videos enhanced by confidence maps unique to ultrasound - referred to as Certainty Weighted Ultrasound Images (CWUS). This allows the decoder to learn relevant local details in noisy ultrasound images.

- Our pre-trained network is trained using ultrasound data of diverse anatomies, including but not limited to 100,000+ images and videos of the heart, liver, lungs, brain, bones, thyroid, nerves., etc., making it the largest pre-trained ultrasound model that is ideally suited for a broad range of ultrasound-related downstream tasks.

## 2. Method

To extend the model for both ultrasound images and videos, we pre-train a single unified model which is built on a Vision Transformer (ViT) (Dosovitskiy et al., 2020). For pre-training, we utilize the basic self-supervised technique of masked auto-encoding (MAE) (He et al., 2022) and build upon it. We use an encoder-decoder architecture where the decoder predicts the Certainty of Weighted Ultrasound Images (CWUS) (see Fig. 2) of the noisy input. The network is trained to minimize the reconstruction loss for the masked pixels of the input ultrasound. Following pretraining, we assess the encoder's performance through transfer learning, while disregarding the decoder. The proposed framework is illustrated in Fig. 1. In the following section, we describe each stage of the pre-training pipeline in detail.

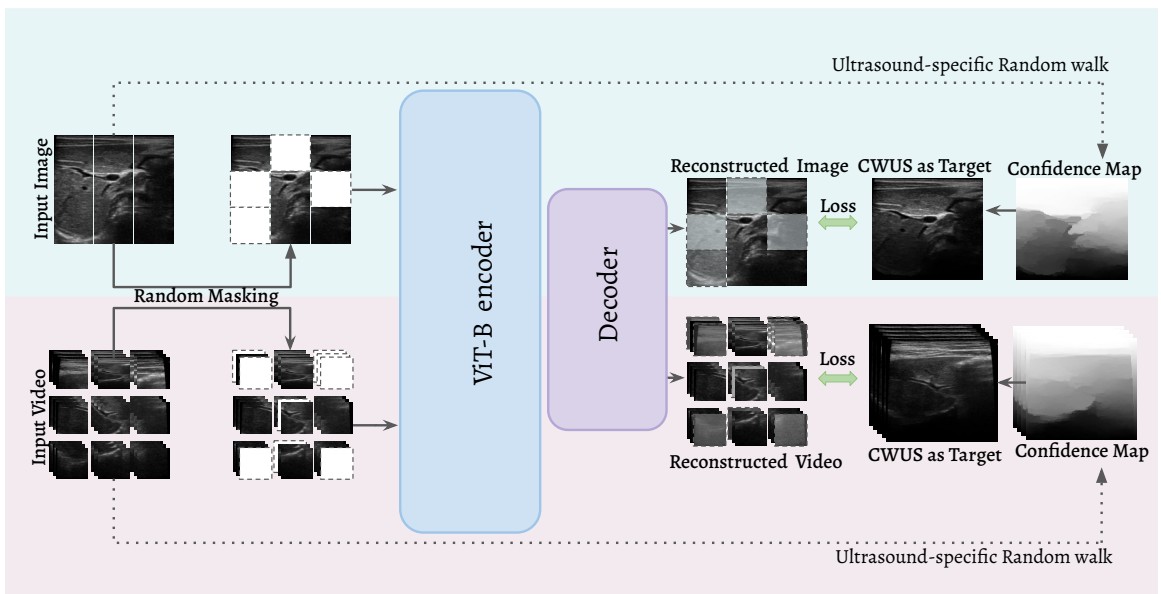

Figure 1: An overview of the proposed UltraMAE pre-training network. The model takes masked spatiotemporal patches of ultrasound images/videos as input and predicts Certainty Weighted Ultrasound Images (CWUS).

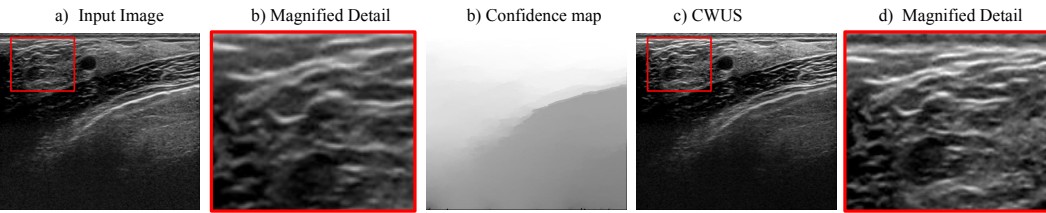

Figure 2: An example of Certainty Weighted Ultrasound Images (CWUS). CWUS images contain reduced noise in uncertain areas.

## 2.1. Certainity Weighted Ultrasound Images (CWUS)

Although there has been a substantial improvement in ultrasound image quality, ultrasound attenuation, and shadowing artifacts are still challenging parts of the image acquisition (Hacihaliloglu, 2017). Various approaches have been discussed in the literature regarding the concept of an ultrasound confidence map. One of these approaches involves using methods that estimate ultrasound attenuation to identify areas that are shadowed or attenuated (Yu and Wang, 2010). By doing so, these methods can provide valuable information about regions in the image that may be unreliable or noisy. We leverage this confidence map to compute a less noisy version of the original ultrasound image, namely Certainty Weighted Ultrasound Images (CWUS). In order to compute CWUS, we construct per-pixel-based confidence maps, which emphasize the uncertainty in attenuated areas of the ultrasound images. (Karamalis et al., 2012). The random walk-based framework with ultrasound domain-specific constraint estimates the probability of a random walk's ability to reach the transducer starting from a pixel, assuming that the likelihood of ultrasound transmission is proportional to the image information (Karamalis et al., 2012). The constraints consider transmission modeling, depth-dependent attenuation, and ultrasound scanline constraints. For the random walk algorithm, the ultrasound image is defined as an undirected weighted graph $G = (V, E)$ with nodes $v \in V$ and edges $E \in e$. The probability of a random walk starting from a pixel to reach the first one of the labeled seeds is calculated using the graph Laplacian matrix as follows,

$$L_{ij} = \begin{cases} d_i \text{ if } i = j \\ -w_{ij} \text{ if } v_i \text{ and } v_j \text{ adjacent nodes} \\ 0 \text{ otherwise.} \end{cases} \tag{1}$$

where $d_i = \sum_j w_{ij}$. The Laplacian matrix can be formulated by utilizing both the graph incidence matrix $A$ and the diagonal matrix of edge weights $C$, resulting in matrix $L = A^T C A$. Then, $L$ is re-ordered and decomposed into blocks of marked nodes $M$ and unmarked nodes $U$ as follows

$$L = \begin{bmatrix} L_M & B \\ B^T & L_U \end{bmatrix}. \tag{2}$$

The final probabilities are then obtained by solving,

$$L_U x_U = -B^T x_M, \tag{3}$$

where, $x_U$ and $x_M$ are the unknown and known probabilities for the unmarked and seed nodes respectively. The attenuation of the signal is defined using the Beer-Lambert law, where we define the attenuated signal $I$ by,

$$I = I_0 exp(-\alpha d), \tag{4}$$

where, $I_0$ is the initial intensity, $\alpha$ is the attenuation coefficient and $d$ is the distance from the source. Ultrasound image formation uses a narrow, focused beam of sound to acquire each scanline. The random walk framework models the beam width, allowing for sound to contribute from structures close to a scanline and limiting bending around reflectors. These

two constraints are integrated into the algorithm by incorporating a new weighting function (Karamalis et al., 2012),

$$
w_{ij} = \begin{cases} w_{ij}^H & \text{if } i,j \text{ adjacent and } e_{ij} \in E_H \\ w_{ij}^V & \text{if } i,j \text{ adjacent and } e_{ij} \in E_V \\ w_{ij}^D & \text{if } i,j \text{ adjacent and } e_{ij} \in E_D \\ 0 & \text{otherwise,} \end{cases}
\tag{5}
$$

where

$$
w_{ij}^H = exp(-\beta(|c_i - c_j| + \gamma)) \tag{6}
$$
$$
w_{ij}^V = exp(-\beta(|c_i - c_j|)) \tag{7}
$$
$$
w_{ij}^H = exp(-\beta(|c_i - c_j| + \sqrt{2}\gamma)) \tag{8}
$$
$$
w_{ij}^H = g_i exp(-\alpha l_i) \tag{9}
$$

and, $E_H$, $E_V$, and $E_D$ are edges along the horizontal, vertical, and diagonal graph direction, $g_i$ is the image intensity respectively. $\alpha$ is the parameter that indicates the likelihood of vertical random walk, $\beta$ affects the randomness and accuracy of the confidence estimation and $\gamma$ penalizes horizontal and diagonal random walks in the graph away from the starting scan line. Finally, we obtain CWUS by multiplying the original input US with the confidence.

### 2.2. UltraMAE

We follow the same pipeline as (Girdhar et al., 2022a) for the network architecture while using ViT-B (Dosovitskiy et al., 2020) as the backbone. The input ultrasound or video is considered as a 4D tensor of $T \times H \times W \times 3$, where $H, W$ represents height and widths and $T$ is the temporal dimension, which is set equal to 1 in the case of images. To process both ultrasound images and videos, an omnivorous (Girdhar et al., 2022b) encoder is utilized that takes variable $N$ spatiotemporal patches. The patches are randomly masked with a factor of $M$ and non-masked patches with positions are fed to the encoder to produce the patch embeddings. Then the output from the encoder is concatenated with $M$ learnable tokens to get $N$ embeddings. The positional encodings to $N$ embeddings are added and fed to the decoder that learns to produce the certainty-weighted ultrasound images. Then, we minimize the $l_2$ distance between the decoder predictions and the CWUS scans over the $M$ masked patches. The transformer decoder consists of 4 layers with 384 embedding dimensions (Girdhar et al., 2022a) that output the CWUS pixels in all input patches.

## 3. Experiments

### 3.1. Datasets

**Pre-training Datasets.** We combine various public and private ultrasound images and videos of various anatomies such as lungs (Chen et al., 2021), bones (Wang et al., 2018; Mohabir, 2020), breasts (Li et al., 2022), heart (Reddy et al., 2023; Ouyang et al., 2020), etc (van den Heuvel et al., 2018). The dataset contains a total of 42,131 ultrasound scans and 63,148 videos. Any annotations or labels for these datasets remain unused during the

pre-training phase. We also don't utilize any pre-training dataset during fine-tuning stages. **Fine-tuning Datasets.** We finetune our model for both classification and segmentation datasets of various human anatomies to demonstrate the generalizability of our pre-trained network. The following datasets are considered in our experiemnts. **i) PCOS Dataset** (Choudhari, 2022): PCOS stands for polycystic ovary syndrome which is a medical condition that affects ovaries. The dataset contains ultrasound images of ovaries with infected and non-infected classes. The training data contains 1,926 images and the test set contains 1,934 images. **ii) Thyroid Nodule Classification** (Yu et al., 2022): Thyroid nodules are growths, either containing solid or liquid material, that develops in the thyroid gland. This dataset contains ultrasound images of thyroids and is classified into three groups- normal, benign, and malignant. The dataset contains 1933 training images and 362 test images. **iii) Hemangioma Dataset** (ShengWang1130, 2021): Hemangioma is a type of birthmark that appears as a raised, rubbery bump with a bright red coloration. It typically develops either at birth or within the first two weeks of life and results from an overgrowth of blood vessels in the skin. The dataset contains 390 training images and 42 test images, and the label contains the segmentation masks of the hemangioma. **iv) Breast Ultrasound Dataset** (Al-Dhabyani et al., 2020): Breast ultrasound uses sound waves to create images of breast structures and helps detect abnormalities such as lumps found during the examination or other diagnostic tests. The dataset contains breast ultrasound images with three classes, benign, malignant, and normal with segmentation masks. The dataset contains 1581 images in total, which is split into 90:10 ratios for training and testing. **v) Thyroid Nodule Segmentation Dataset** (Pedraza et al., 2015): The dataset contains the ultrasound images and their corresponding segmentation masks of thyroid nodules. The dataset contains 870 images in total and is split into 90:10 ratios for training and testing.

### 3.2. Implementation Details

The proposed framework is implemented using PyTorch. We utilize the ViT-B backbone without [CLS] token with an omnivorous encoder. Both the encoder and the decoder utilize sinusoidal positional encoding for the patches. For pre-training, we mask out 90% and 95% of patches respectively in images and videos, as this ratio has been found to yield optimal results (Girdhar et al., 2022a). To process video ultrasound, we capture a segment of T = 16 frames with a frame rate of 6 FPS. The dimensions of both input images and videos are resized into $224 \times 224$ pixels. The rest of the parameters for the pre-trained model are the same as (Girdhar et al., 2022a). The model is trained for 800 epochs on the pretraining datasets. For the pre-training process, we utilized 4 NVIDIA RTX A5000 GPUs, each with a capacity of 24 GB. For the downstream classification task, we discard the original decoder and add a linear layer of 768 input dimesions (Girdhar et al., 2022a) which maps this feature vector to a number of classes-dimensional output vector. For the downstream segmentation task, we utilize the same decoder architecture as pre-trained network, except the last layer produces pixel-wise probabilities of the binary segmentation masks.

Table 1: Evaluation results for the fine-tuning. The networks are pre-trained using both images and videos of ultrasounds. Boldface numbers indicate the best classification/segmentation performance. The number indicates accuracy for classification and the IoU score for segmentation.

| | Classification | | | Segmentation | | |
|---|---|---|---|---|---|---|
| Method | PCOS (Choudhari, 2022) | BUSI (Al-Dhabyani et al., 2020) | TN-C (Yu et al., 2022) | HEM (ShengWang1130, 2021) | BUSI (Al-Dhabyani et al., 2020) | TN-S (Pedraza et al., 2015) |
| USCL (Chen et al., 2021) | 0.9812 | 0.8551 | 0.8352 | 0.9080 | 0.8121 | 0.6553 |
| SUSCL (Basu et al., 2022) | 0.9835 | 0.8465 | 0.8494 | 0.9095 | 0.8180 | 0.6598 |
| MAE (He et al., 2022) | 0.9885 | 0.8579 | 0.8551 | 0.9106 | 0.8223 | 0.6626 |
| UltraMAE (Ours) | **0.9968** | **0.8721** | **0.8693** | **0.9275** | **0.8382** | **0.6843** |

### 3.3. Comparison with Previous Literature

In this section, we compare UltraMAE with previous self-supervised pre-training methods. The results are shown in Table 1. We pre-train a vanilla masked-autoencoder with the pre-train dataset and consider the model as a baseline for the masking-based pre-training method. Additionally, we compare our pre-training framework with USCL (Chen et al., 2021) and SUSCL (Basu et al., 2022), which are specialized networks for pretraining on US datasets and are the current state-of-the-art. We chose USCL and SUSCL because they are ultrasound-specific pre-training models. We can observe that UltraMAE outperforms the previous state-of-the-art pretrained network as the model utilizes a masking-based approach. It's also evident that UltraMAE outperforms vanilla MAE, as the network leverages an omnivorous encoder that captures better data representation.

### 3.4. Ablation Studies

In this section, we report the fine-tuning results on all the datasets to validate the different contributions of our proposed approach. We pre-train our backbone network in five different settings as shown in Table 2. The network trained on both videos and images performed better than the network trained on a single modality. Simply turning all the video into image frames and applying MAE result in a lot of redundancy, hence the performance is not on par with the combined model. The performance improves even more across all the datasets when the UltraMAE utilizes the confidence map for reconstruction, as adding a deblurring component has proven to help while reconstructing masked tokens (Kang et al., 2023).

### 3.5. Limited Data Experiments

In our study, we conducted a series of experiments on various splits of the dataset to demonstrate the effectiveness of our pretrained method under limited-annotation scenarios. Specifically, we performed experiments on PCOS (classification) and HEM (segmentation) data across four different data splits from 10% to 50% and visualized the results in Fig. 3. The figures indicate that the gap between the results of UltraMAE and the from-scratch experiments is particularly pronounced when working with limited amounts of data. This emphasizes the critical importance of pre-training for ultrasound imaging tasks, where data is often scarce.

Table 2: Ablations results. Boldface numbers indicate the best classification/segmentation performance. The number indicates accuracy for classification and the IoU score for segmentation.

| | Classification | | | Segmentation | | |
|---|---|---|---|---|---|---|
| Method | PCOS (Choudhari, 2022) | BUSI (Al-Dhabyani et al., 2020) | TN-C (Yu et al., 2022) | HEM (ShengWang1130, 2021) | BUSI (Al-Dhabyani et al., 2020) | TN-S (Pedraza et al., 2015) |
| Only Images (He et al., 2022) | 0.9885 | 0.8579 | 0.8551 | 0.9106 | 0.8223 | 0.6626 |
| Only Videos (Tong et al., 2022) | 0.9762 | 0.8494 | 0.8323 | 0.9113 | 0.8102 | 0.6275 |
| Video + Images (Girdhar et al., 2022a) | 0.9812 | 0.8664 | 0.8465 | 0.9170 | 0.8201 | 0.6626 |
| UltraMAE (Ours) | **0.9968** | **0.8721** | **0.8693** | **0.9275** | **0.8382** | **0.6843** |

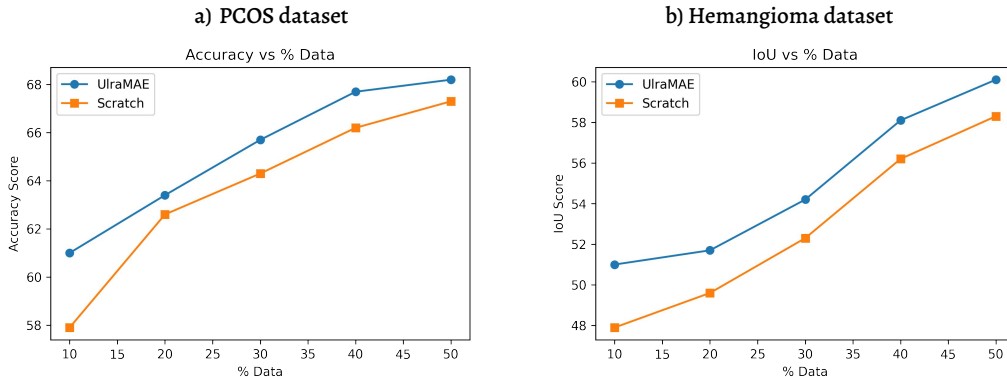

Figure 3: The impact of training on reduced data. It can be observed that UltraMAE pre-training on reduced training data significantly improves the overall performance across both tasks.

## 4. Conclusion

In this work, we have introduced the first unified masking-based pre-training model for the ultrasound domain. To the best of our knowledge, this is the first study that leverages confidence-guided enhanced videos and images of ultrasound for masked pertaining. We utilized video modalities as well, as there are a lot more video ultrasound datasets but not utilized during the pre-training strategy. We train our model with a wide range of ultrasound images and videos of various anatomies obtained using different types of machines, making it the largest ultrasound pre-trained model. We evaluate our model for both classification and segmentation tasks of different human anatomy. In the future, we aim to train even larger models with other ultrasound modalities such as 3D ultrasounds. Lastly, there is also scope for improvement by making $\alpha, \beta$ and $\gamma$ trainable parameters for constructing optimal CWUS images.

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
