# OpenReview forum: "UltraMAE: Multi-modal Masked Autoencoder for Ultrasound Pre-training"
_MIDL.io/2024/Conference — MIDL 2024 Poster_

### Official Review · Reviewer_jtaL · 2024-02-19

**Confidence:** 3
**Preliminary Rating:** 4
**Recommendation:** Poster
**Final Rating:** 4

**Summary:**

This paper proposes a novel method for ultrasound image analysis that utilizes an ultrasound-specific confidence map to guide low-level representation learning through masked feature acquisition. The authors demonstrated the efficacy and versatility of the framework in tackling both classification and segmentation tasks.

**Strengths:**

1. Introducing an innovative framework designed to enhance the quality of ultrasound images/videos through the incorporation of confidence maps specific to ultrasound imaging.

2. The pre-trained network undergoes training with a diverse range of ultrasound data, ideally suited for various ultrasound-related tasks downstream.

**Weaknesses:**

1. The author's assertion regarding unifying image and video within a pretraining framework appears exaggerated. Image pretraining, essentially a subset of video training, has been previously suggested in works such as USCL or VideoMAE.

2. The question remains open regarding whether a model pretrained on both modalities exhibits superior performance compared to a model pretrained on just one. Clarification is essential regarding whether the quantity of training frames is controlled, as it significantly impacts determining whether improvements stem from the quantity of training samples or solely from the temporal dependencies introduced by video. The author should address this ambiguity.

**Detailed Comments:**

1. Bigger font in Table 1/2's font is suggested

**Justification Of Final Rating:**

I am a bit disappointed by the absence of a rebuttal for my concern, as it suggests a missed opportunity for clarification. However, considering the merit of the work, I maintain my final rating as a weak accept.

**Justification Of The Preliminary Rating:**

The presentation of the work is generally easy to follow and the improvement is also substantial, subject to clarification on the quantity of training frames between different frameworks. The methodology is very simple yet effective.

**Questions To Address In The Rebuttal:**

see weakness

**Special Issue:**

No

---

### Official Review · Reviewer_TfEs · 2024-02-26

**Confidence:** 4
**Preliminary Rating:** 3
**Recommendation:** Poster

**Summary:**

This paper proposes a strategy for developing a pre-trained network for ultrasound images and videos. Their model is built on a Vision Transformer (ViT). For pre-training, they utilize the self-supervised technique of masked auto-encoding (MAE). They propose to reconstruct the certainty of weighted ultrasound images leveraging confidence maps to guide low-level representation learning. The model is tested for both classification and segmentation across a range of pathologies.

**Strengths:**

- Paper is well written, easy to follow and a complete overview of related work is provided
- Figure 1 provides a good overview of the method.
- Unified method that integrates both ultrasound images and videos leveraging a transformer.
- Model for diverse anatomies: heart, liver, lungs, brain, bones, thyroid, etc.
- The authors propose to reconstruct certainty weighted ultrasound, instead of the original image, to recover details and be more robust to noise.
- Ablation study to validate the different contributions of their method, BUT performance is still reported for a single model.

**Weaknesses:**

- As far as I understand, reported results in Table 1 are for a single model. UltraMAE outperforms all the compared methods by a small improvement. I am concerned about the lack of cross-validation and the statistical significance of the improvement.
- The citation for the BUSI dataset references a Kaggle dataset. Could you please provide the source of this dataset? There is an open discussion regarding image duplication within both the benign and malignant classes of this dataset. I'm concerned about the reliability of the results obtained from using this dataset.
- Regarding the experiments about limited data. I expect any pre-training strategy to work better than the from-scratch experiments. A fairer comparison would be against other pre-training strategies like USCL or SUSCL, or pre-training datasets like ImageNet.
- In the paper it is stated that the code and model will be made available after the review process. However, as a reviewer I cannot ensure the reproducibility and transparency of the results at this stage.
- There are some important details missing in the abstract, e.g. about the proportion of “limited data” that makes a difference for the pre-training, or the use of “efficacy” in terms of training time or accuracy?

**Detailed Comments:**

Minor improvements:
- I don’t see a difference between Fig. 2 (b) and (f). I would suggest highlighting in the figure or caption or in the text what this figure is showing.
- Missing space between 0 and text in Eq. (1)
- Matrix B is not defined in Eq. (2)
- Definition of some parameters is missing in Eqs. (6-9), e.g. g_i is not defined in Eq. (9)
- $\alpha$ is defined to be related to vertical random walks, but used in $w_{ij}^H$ Eq. (9), is this correct?
- In Section 3.3 I would include the full name of the compared methods (USCL, SUSCL) for easier readability
- Implementation details. This model is trained for 800 epochs. How many GPUs were used for training?
- There are no visual results of the segmentation. I would suggest including a figure in Supplementary material at least.

**Justification Of The Preliminary Rating:**

This paper presents a strong but complex method.  I am hesitant about the necessity of employing such complexity to achieve only a slight improvement. My main concerns are about the validation, in terms of the statistical significance of the reported improvement, and the use of a dataset (BUSI) that is missing the provenance and any documentation.

**Questions To Address In The Rebuttal:**

1. Concerns about validation (Table 1 and Table 2)
- Justify why there is no cross-validation in this study (Table 1) and why performance is reported for a single model (Table 1 and Table 2)
- MAE seems to be a much simpler model, is it worth it adding the extra complexity? - elaborate

2. Concerns about BUSI dataset source
- Could you specify the source of this dataset and the number of images for the benign and malignant classes?
- There's an open discussion regarding potential image duplication within both the benign and malignant classes. I'm concerned about the reliability of the results obtained from using this dataset, particularly due to ethical implications surrounding the use of research datasets with unknown sources and lack of documentation.

3. Limited data experiment
I expect any pre-training strategy to work better than the from-scratch experiments. A fairer comparison would be against other pre-training strategies like USCL or SUSCL, or pre-training datasets like ImageNet.
- Could you justify the choice of baselines?
- How about comparing to other natural images datasets such as ImageNet?

**Special Issue:**

No

---

> ### Author Response · Authors · 2024-03-12
>
> Thank you for your insightful review. Regarding cross-validation and statistical significance, we acknowledge the importance of specifying the p-value, which is less than 0.05. We plan to update the paper with mean and standard deviation in future versions.
>
> Regarding the BUSI dataset source and issue: Al-Dhabyani, W., Gomaa, M., Khaled, H., & Fahmy, A. (2020) presented a dataset of breast ultrasound images in their work titled "Dataset of breast ultrasound images" published in Data in Brief, volume 28, article 104863. We appreciate you bringing to our attention the issue of data duplication across both malignant and benign classes. We will remove the duplicated instances. However, it's important to note that we used the BUSI dataset solely for tumor segmentation purposes (binary segmentation), which means the presence of duplicated malignant/benign images does not affect the outcome of the diagnosis. This is unless they appear in both the training and test sets, which would indeed constitute data leakage—a separate concern. We will take steps to identify and remove any such instances. Thank you for highlighting this issue.
>
> Regarding the clarification on pre-training: In the context of limited data settings, we have not compared our approach with other baselines since we have already conducted comparisons using the full dataset scenario. The objective of that experiment was to demonstrate that pre-training yields better results than training from scratch, especially when the available data is extremely limited. Nonetheless, we recognize that any network benefiting from pre-training is likely to outperform a network trained from scratch. Both the USCL and SUSCL methodologies were compared with ImageNet pre-training; thus, we did not train any model explicitly using an ImageNet model. We chose USCL and SUSCL because they are ultrasound-specific pre-training models. However, for the PCOS dataset, we used a ResNet-50 model pre-trained on ImageNet and fine-tuned it to achieve a 98.12% accuracy with the full dataset. We will clear and update it in future versions.
>
> Minor improvements include the following updates and clarifications:
>
> 1. Figure 2 will be revised to include better-quality images that more clearly highlight the differences. Additionally, other editorial oversights such as missing spaces and the need to define certain matrices will be corrected.
>
> 2. Regarding Equation 9, we confirm its correctness. To prevent any confusion, we will amend the equation in the final version to read "g_i * exp..."
>
> 3. For the pre-training process, we utilized 4 GPUs, each with a capacity of 24 GB.
>
> 4. We plan to include segmentation results in a future version of the paper to provide a more comprehensive analysis.
>
> 5. Baselines are chosen as they are specifically designed for ultrasound modalities, as well as, already compared against other pre-training strategies. Moreover, we also only train on MAEs/VideoMAEs for fair comparison with MAE-based models.
>
> MAE seems to be a much simpler model, is it worth it adding the extra complexity?
>
> We utilized video modalities as well, as there are a lot more video ultrasound datasets but not utilized during the pre-training strategy. Simply turning all the video into image frames and applying MAE will result in a lot of redundancy. Learning from both images and spatiotemporal data proved to improve [1] the overall performance, and, adding a deblurring component helps [2] while reconstructing masked tokens. We try to combine both for a powerful versatile ultrasound-based foundation model that can take both image and video input.
>
> Girdhar, R., El-Nouby, A., Singh, M., Alwala, K. V., Joulin, A., & Misra, I. (2023). Omnimae: Single model masked pretraining on images and videos. In Proceedings of the IEEE/CVF conference on computer vision and pattern recognition (pp. 10406-10417).
> Kang, Q., Gao, J., Li, K., & Lao, Q. (2023, October). Deblurring masked autoencoder is better recipe for ultrasound image recognition. In International Conference on Medical Image Computing and Computer-Assisted Intervention (pp. 352-362). Cham: Springer Nature Switzerland.

---

> > ### Author Response · Authors · 2024-03-26
> >
> > We believe that we have thoroughly addressed the reviewer's concerns and kindly request the reviewer to let us of any additional issues following our response. We thank the reviewer for their time.

---

> > ### Comment · Reviewer_TfEs · 2024-03-26
> >
> > Thanks to the authors for answering my questions.
> >
> > 1. BUSI dataset source
> > I appreciate the clarification. However, in the manuscript, the reference for the BUSI dataset is still a Kaggle URL. As I pointed out in my original review, the dataset hosted on Kaggle doesn't contain information about the provenance of the BUSI dataset, nor does it include any documentation. Regarding the duplicated malignant/benign images, could you clarify if you have taken any steps to identify and remove such instances?
> >
> > 2. Pre-training
> > Thanks for the explanation. Unfortunately, the changes or additions haven't been highlighted or colored, so this makes it difficult to review them. Could you clarify the changes in the manuscript about the pre-training strategies and ImageNet?
> >
> > 3. Comparison with MAE. Thanks for the clarification about the difference with respect to MAE. However, I am not convinced about the argument of "powerful versatile foundation model".

---

### Official Review · Reviewer_4c6e · 2024-03-05

**Confidence:** 4
**Preliminary Rating:** 3
**Final Rating:** 2

**Summary:**

The paper proposes pretraining masked autoencoder where: i) both ultrasound images (US) and videos are utilized leveraging the transformer-based omnivore architecture, and ii) the reconstruction target for the decoder is certainty weighted ultrasound image which is obtained by weighting original US image with a confidence map generated using random walk from each pixel guided by US image formation model. Pretraining is performed using large number of images and videos, and then classification and segmentation performance is evaluated by finetuning in a set of separate small-scale datasets. The results show improvement over few other self-supervised pretraining.

**Strengths:**

- The paper is written well and easy to understand
- Deep Networks pretrained on US imaging modality specific models with large scale US image and video data  can be useful for diverse tasks.
- Finding the right decoder targets in masked autoencoder, inspired by knowledge of US image formation model could be a reasonably good direction (but requires stronger validation)

**Weaknesses:**

- One of the major novel contributions of the paper seem to be in using CWUS as the decoder's target in masked autoencoder training. However, there is limited ablation study on its role against various baselines: keeping same training data and test sets, and same transformer-based omnivore architecture with masked autoencoding, but different decoder targets (noisy version of input, random weighting, edge-based saliency weighting that does not necessarily take US physics into account etc.)

- Some basic baselines and ablation studies could have provided better insights into the method's ability to learn right representation specific to US images (detailed comments)

**Detailed Comments:**

- A very basic baseline in Table 2 during ablation would be to finetune the same architecture and of same size with random initialization instead of the proposed pertaining; how much improvement for each test dataset separately is being added by pertaining?
- While the two SOTA in US are reasonable for comparison, are they trained and tested in the same sets of data for SOTA in original version? If not, adding a couple of most popular SSLs such as SIMCLR in the current setting would provide a better idea regarding the value of MAE; this will also support the Introduction, which mentioned that Masked autoencoder architecture "shown to outperform", although no work is cited to support this where the experiments are done in settings similar to the one in this paper.
- Are USCL and SUSCL using the same backbone and architecture size?
- Fig 2 caption: reduced noise in uncertain or certain areas?

**Justification Of Final Rating:**

The main concern persists as the rebuttal keeps this for future work: lack of proper ablation studies to identify whether the improved performance comes from the key contribution of the paper, ultrasound physics-based reconstruction target, or not.

**Justification Of The Preliminary Rating:**

While the paper proposes interesting idea and combines a large number of data to create US-specific pretrained model, limited baseline comparison provides ample room for improvement in experiments that will provide better insights into the utility of the proposed method.

**Questions To Address In The Rebuttal:**

Questions in Weakness and detailed comments.

---

> ### Author Response · Authors · 2024-03-18
>
> Thank you for your review.
>
> Q: One of the major novel contributions of the paper seem to be in using CWUS as the decoder's target in masked autoencoder training. However, there is limited ablation study on its role against various baselines: keeping same training data and test sets, and same transformer-based omnivore architecture with masked autoencoding, but different decoder targets (noisy version of input, random weighting, edge-based saliency weighting that does not necessarily take US physics into account etc.)
>
> A: Regarding the ablation studies, while Table 2 provides insights into the effects of different network architectures, we did not explore alternative reconstruction methods beyond those guided by ultrasound (US) physics. Although existing literature suggests that approaches such as input blurring can enhance feature learning, our focus was solely on US-based reconstruction techniques. We acknowledge the potential benefits of exploring these alternative methods and plan to incorporate them in future research efforts.
>
>
>
> Q: A very basic baseline in Table 2 during ablation would be to finetune the same architecture and of same size with random initialization instead of the proposed pertaining; how much improvement for each test dataset separately is being added by pertaining?
>
> A: For the initialization process, Section 3.3 and Figure 3 describe our approach, which involves randomly initializing the network while maintaining the same architecture across all experiments.
>
> Q: While the two SOTA in US are reasonable for comparison, are they trained and tested in the same sets of data for SOTA in original version? If not, adding a couple of most popular SSLs such as SIMCLR in the current setting would provide a better idea regarding the value of MAE; this will also support the Introduction, which mentioned that Masked autoencoder architecture "shown to outperform", although no work is cited to support this where the experiments are done in settings similar to the one in this paper.
>
> A: Regarding benchmarking against state-of-the-art (SOTA) models, we ensured that all baselines were trained and evaluated using the identical dataset as those considered SOTA. Given that one of the referenced papers already includes a comparison with SimCLR and demonstrates superior performance of the baselines, we chose not to replicate these experiments.

---

> > ### Author Response · Authors · 2024-03-26
> >
> > We believe that we have thoroughly addressed the reviewer's concerns and kindly request the reviewer to let us of any additional issues following our response. We thank the reviewer for their time.

---

> > ### Comment · Reviewer_4c6e · 2024-03-28
> > **Assessing the value of CWUS**
> >
> > The key contribution of the paper is the approach claimed to be guided by US physics (CWUS), rather than the autoencoder architectures; thus, the main question is what value is CWUS bringing compared to other commonly used reconstruction targets. Hence, the answer that "the focus was solely on US-based reconstruction techniques" does not address the main concern I have with the paper.

---

### Official Review · Reviewer_mRLk · 2024-03-06

**Confidence:** 4
**Preliminary Rating:** 3
**Final Rating:** 3.5

**Summary:**

This paper introduces a unified pre-trained model for large-scale unlabeled ultrasound datasets, comprising both images and videos. The contributions are threefold. Firstly, instead of reconstructing raw pixels like MAE, the paper argues that the Certainty of Weighted Ultrasound Images is a more suitable target for reconstruction, particularly due to the potential noise in ultrasound images. Secondly, the paper pioneers pre-training on a large scale of 100,000 videos and images, demonstrating enhanced generalization. Lastly, the paper validates the effectiveness of segmentation and classification tasks.

**Strengths:**

- The paper integrates multi-modal data, such as images and videos, to enhance the understanding of ultrasound images, which is significant for the community.
- It is noteworthy that the reconstruction target is adjusted for ultrasound data with significant noise.
- Pre-training on large-scale unlabeled datasets is conducted for representation learning.
- The paper demonstrates the effectiveness of multiple datasets on two tasks.

**Weaknesses:**

- In the methodology section, it appears that videos are processed similarly to images by splitting them into individual frames and feeding each frame to the backbone. This design overlooks temporal correspondence in self-supervised learning for video analysis[1,2].
- In the experiments:
    - VideoMAE[3] could serve as a strong baseline when pre-training solely with video data. The proposed method should also be compared with UniMISS[5], which deals with 2D and 3D data together.
    - Could the author evaluate some video-related tasks? Since the proposed method is pre-trained on both images and videos, it should generalize to these modalities in downstream tasks.
    - Could the author supplement the performance metrics when scaling the pre-training datasets? It would be intriguing to observe if scaling laws[4] also apply to ultrasound images.
    - Can the author provide performance metrics when varying the mask ratio?

[1] X. Wang, A. Jabri, and A. A. Efros, “Learning Correspondence From the Cycle-Consistency of Time,” in CVPR, Long Beach, CA, USA, Jun. 2019, pp. 2561–2571. [2] H. Duan, N. Zhao, K. Chen, and D. Lin, “TransRank: Self-supervised Video Representation Learning via Ranking-based Transformation Recognition,” in _CVPR_, New Orleans, LA, USA, Jun. 2022, pp. 2990–3000.
[3] Z. Tong, Y. Song, J. Wang, and L. Wang, “VideoMAE: Masked Autoencoders are Data-Efficient Learners for Self-Supervised Video Pre-Training,” in _NeurIPS_, 2022.
[4] J. Kaplan _et al._, “Scaling Laws for Neural Language Models.” arXiv, Jan. 22, 2020.
[5] Y. Xie, J. Zhang, Y. Xia, and Q. Wu, “UniMiSS: Universal Medical Self-supervised Learning via Breaking Dimensionality Barrier,” in _ECCV_. 2022, pp. 558–575.

**Detailed Comments:**

See weakness

**Justification Of Final Rating:**

I appreciate the authors' efforts in addressing the concerns raised in the rebuttal and I would like to express my gratitude for their time and dedication.

While there are still some unresolved concerns that remain after reviewing the rebuttal, I am pleased to see this paper accepted by MIDL.

**Justification Of The Preliminary Rating:**

- The research question addressed in this paper holds significant relevance for the community, shedding light on the efficacy of harnessing large-scale multi-modal data to enhance ultrasound-related tasks.
- The discovery regarding the reconstructions of Certainty Weighted Ultrasound Images signifies a potentially more suitable approach.
- Nevertheless, it's imperative to acknowledge that the proposed method overlooks several intrinsic characteristics of videos.
- Additional experiments are necessary to supplement,  see weakness parts.

**Questions To Address In The Rebuttal:**

- For the experiments:
    - VideoMAE[3] could serve as a strong baseline when pre-training solely with video data. The proposed method should also be compared with UniMISS[5], which deals with 2D and 3D data together.
    - Could the author evaluate some video-related tasks? Since the proposed method is pre-trained on both images and videos, it should generalize to these modalities in downstream tasks.
    - Could the author supplement the performance metrics when scaling the pre-training datasets? It would be intriguing to observe if scaling laws[4] also apply to ultrasound images.
    - Can the author provide performance metrics when varying the mask ratio?

---

> ### Author Response · Authors · 2024-03-18
>
> Thank you for your reviews:
> Q: VideoMAE[3] could serve as a strong baseline when pre-training solely with video data. The proposed method should also be compared with UniMISS[5], which deals with 2D and 3D data together.
>
> Table 2 specifies that VideoMAE is utilized solely for video data. We will consider UniMISS in future research effort.
>
> Q: Could the author evaluate some video-related tasks? Since the proposed method is pre-trained on both images and videos, it should generalize to these modalities in downstream tasks.
>
> At present, we lack access to a video-related downstream dataset suitable for our experiments; however, we are keen on incorporating such datasets in our future research.
>
> Q: Can the author provide performance metrics when varying the mask ratio?
>
> The selection of the masking ratio was guided by the parameters used in the foundational work of OmniMAE, where a 90-95% masking ratio for both images and videos was found to yield optimal results. Therefore, we adhered to these established percentages in our study.

---

> > ### Author Response · Authors · 2024-03-26
> >
> > We believe that we have thoroughly addressed the reviewer's concerns and kindly request the reviewer to let us of any additional issues following our response. We thank the reviewer for their time.

---

### Meta-Review · Area_Chair_meKw · 2024-04-03

**Recommendation:** Accept (Poster)
**Confidence:** 4

**Metareview:**

This paper introduces a unified pre-trained model for large-scale unlabelled ultrasound datasets from diverse anatomies. Initially reviewers raise diverse comments ( three borderline and one weak accept in preliminary rating). After response period, one reviewer raised the rating to borderline accept and one decreased the rating to weak reject. Reviewers appreciate the merits of this paper on pre-training on large-scale unlabelled ultrasound dataset, reconstructing certainty weighted ultrasound, instead of the original image, to recover details and be more robust to noise and demonstrating better performance, etc. While reviewers also mention some key issues remain unsolved, including clarification is essential regarding whether the quantity of training frames is controlled, the unclear value of CWUS bringing compared to other commonly used reconstruction targets, etc. After reconciling all the comments, AC tends to suggest the decision of accept given the merits of this paper. In the meanwhile, AC strongly suggest the authors take reviewers’ comments into thorough revision for preparing the final version.

---

### Decision · Program_Chairs · 2024-04-06

Accept (Poster)